# The efficiency in the ordinary hospital bed management in Italy: An in-depth analysis of intensive care unit in the areas affected by COVID-19 before the outbreak

**Fabrizio Pecoraro**[1]*, **Fabrizio Clemente**[2], **Daniela Luzi**[1]

**1** Institute for Research on Population and Social Policies, National Research Council, Rome, Italy,
**2** Institute of Crystallography, National Research Council, Monterotondo (Rome), Italy

* f.pecoraro@irpps.cnr.it

**Data Availability Statement:** The excel file is available from the Zenodo database. doi: 10.5281/zenodo.3339345.

## Abstract

Since the end of February 2020 a severe diffusion of COVID-19 has affected Italy and in particular its northern regions, resulting in a high demand of hospitalizations in particular in the intensive care units (ICUs). Hospitals are suffering the high degree of patients to be treated for respiratory diseases and the majority of the health structures, especially in the north of Italy, are or are at risk of saturation. Therefore, the question whether and to what extent the reduction of hospital beds occurred in the past years has biased the management of the emergency has come to the front in the public debate. In our opinion, to start a robust analysis it is necessary to consider the Italian health system capacity prior to the emergency. Therefore, the aim of this study is to analyse the availability of hospital beds across the country as well as to determine their management in terms of complexity and performance of cases treated at regional level. The results of this study underlines that, despite the reduction of beds for the majority of the hospital wards, ICUs availabilities did not change between 2010 and 2017. Moreover, this study confirms that the majority of the Italian regions have a routinely efficient management of their facilities allowing hospitals to treat patients without the risk of having an overabundance of patients and a scarcity of beds. In fact, this analysis shows that, in normal situations, the management of hospital and ICU beds has no critical levels.

## Introduction

The number of individuals infected with severe acute respiratory syndrome coronavirus 2 (SARS-CoV-2), the virus causing COVID-19 emergency, is dramatically increasing worldwide [1]. The first person-to-person transmission in Italy was reported on February 21st, 2020, and led to an infection chain that represents the largest COVID-19 outbreak outside Asia to date. As of 29th of March 2020, Italy is the second most affected country in the World and the first in Europe, with more than 97.000 confirmed cases according to the Italian Department of Civil Protection [2]. What is still an open question is the reason why the spread of this

**Funding:** The author(s) received no specific funding for this work.

**Competing interests:** The authors have declared that no competing interests exist.

epidemic has had its most virulent peak in the northern regions of the country [3] leading to a high level of both mortality and fatality rates in comparison to other countries [4, 5] as well as a high contagion and mortality of healthcare workers both in the primary and hospital care [6]. Currently, the 87% of the confirmed actual cases are inhabitants of the northern regions that counts the 55% of the total Italian population. Among them, Lombardia, where the first cases were identified at the end February, is still the region with the majority of individuals infected by the COVID-19 virus. Similar results can be noted considering the hospitalization in general (88% in the northern regions and 42% in Lombardia) and the intensive care units in particular (85% in the northern regions and 35% in Lombardia). This data is provided and daily updated by the Italian Civil Protection Department [7]. These numbers are indicative of the spread of the infection and underline that one of the main challenges to be faced by the National and the Regional Health Systems is the management of hospital resources across the country in terms of professionals, medical devices and hospital beds, with a particular attention on the intensive care unit (ICU) [8]. Given this high demand of hospitalization and critical care, Italian hospitals and in particular those located in the northern regions, have been overloaded. The majority of them are struggling to cope with patients infected by the COVID-19 in addition to those who are hospitalized for other diseases. This has led the actual and past national and regional governments to be heavily criticized for having reduced the number of beds in the past years, in particular those located in the ICUs [9–11]. Different newspapers have highlighted that a reduction in the number of beds has affected many hospitals over the territory. This resource reduction was mainly related to the progressive cutting of the national and local budget in the last decade that led the regional health services to close a significant number of local small dimension hospitals that generally have higher costs [12]. This lack of beds was confirmed by a recent study of the OECD (Organisation for Economic Co-operation and Development) that reported that our country counts 2.6 hospital beds per 1.000 inhabitants [13], ranking Italy at the 19th place over 23 countries with Germany having more than 6 beds per 1.000 inhabitants. However, this data does not specifically consider the ICU. A most recent study reporting the number of beds for the ICU was published by Rhodes et al. in 2012 [14]. Also in this case the number of beds located in Italy was below the European average with 12.9 beds per 100.000 inhabitants, while Germany was equipped with 29.2 beds. The analysis of the Italian National Healthcare Service should consider that health services are organized and delivered under the responsibility of local authorities structured at a regional level, while the Italian Government has a weak strategic leadership [15, 16].

Starting from these premises, the aim of this study is to analyse the Italian regional hospital systems before the outbreak of the COVID-19, in order to assess the efficiency and the performance of the hospital bed management in the past years, so to indicate a reference point for future analysis. Moreover, this study may help to determine if the reduction of hospital resources may have an impact on the functioning of a hospital and on the efficiency in the management of clinical cases [17]. Specific attention is given to the management of the ICU departments. In this perspective, there are two methodologies adopted to assess the efficiency of a health structure in the management of clinical hospitalized cases. The first one is the hospital bed management [18–20] that provides an overall description of the use of beds by health structures. The second methodology evaluates the performance of a hospital considering the complexity of cases treated by the structure [21, 22]. Both methodologies investigate hospital performances providing a helpful snapshot for healthcare managers for the evaluation of healthcare systems [23].

The paper is structured as follows: after the materials and methods paragraph, a brief description of the virus diffusion and workload of the hospital infrastructures is reported on the basis of data available so far to provide an overview of the extraordinary efforts of the

Italian health and social care professionals. After that, a comparison of resources availability across the Italian regions is performed to analyse the level of beds reduction in each region. For this analysis we adopted the data captured in the years 2010 and 2017 which are the first and the last available data exposed yearly in the Ministry of Health website [24]. Finally, the results of the hospital bed management analysis are reported to capture the efficiency as well as the complexity and performance of patient hospitalizations.

## Materials and methods

### Data

This paper is focused on two main data sources. The first one [24] provides daily data on the number of individuals affected by the COVID-19 as well as patients hospitalized in the ward and specifically in the ICUs during the outbreak. This website exposes official and continuously updated information produced by the Italian Civil Protection Department and adopted by the Ministry of Health for its daily bulletin [25]. The second information flow [24] concerns data on hospital bed management. Administrative and clinical data produced during the hospitalization process are collected in Discharge Report Forms and sent by each hospital information system to be centrally gathered by the Ministry of Health. They describe each service provided to a patient as well as facilities available and staff employed in the relevant structure. This information is aggregated and published yearly in the web site of the Ministry of Health [24]. Data analysed in this study refers to the year 2017, the latest most outdated information currently published by the Ministry of Health. Moreover, to examine the differences across years in terms of resources availability and their adoption the analysis has been carried out also on data captured for the year 2010. This study considers both the public and private institutes, excluding the private nursing homes. The datasets gathered from the Ministry of Health website and the relevant analyses performed during the current study are available in the Zenodo repository [26].

### Methodologies

The overall description of the hospital bed management is assessed using indicators computed on the basis of the number of beds, the patient discharged over a specific period of time (i.e. a year) and the total number of inpatient days (i.e. the overall number of hospitalization days of all patients) [27]. In particular, the indicators used are defined as follows:

• Beds Occupancy Rate (BOR): percentage of inpatient beds occupied over a specific period;

• Average Length Of Stay (AvLOS): average number of days that an inpatients remained in the hospital;

• Turnover Interval (TOI): number of days in which an available bed remains empty between the discharge of a patient and the admission of a next one;

• Beds Turn Over (BTO): average number of patients "passing through" each bed during a specific period.

We also adopted reference thresholds for TOI (1<TOI<3) and BOR (85%>BOR>75%) [28–31] to classify each region in the following areas: 1) the red one that identifies regions where both TOI and BOR are outside the reference threshold; 2) the yellow area that reports regions where either TOI or BOR are outside the threshold and 3) the green area that identifies regions where both indicators are within the reference thresholds.

The complexity and the performance of each clinical department in the management of clinical hospitalization cases are described, respectively, by the Case Mix (CMI) and the

Performance Index (PI). In particular, the former indicates the degree of complexity of clinical cases treated in each hospital with respect to a benchmark level (i.e. the national average), while the PI compares the performance of the observed hospital considering the inpatient length of stays compared to the benchmark level.

These indicators are computed according to the following formulas:

$$CMI = \frac{\sum_{j=1}^{n} AvLOS_j^{REF}\left(\frac{d_j^{STR}}{d_{ALL}^{STR}}\right)}{AvLOS_{ALL}^{REF}} \tag{1}$$

$$PI = \frac{\sum_{j=1}^{n} AvLOS_j^{STR}\left(\frac{d_j^{REF}}{d_{ALL}^{REF}}\right)}{AvLOS_{ALL}^{REF}} \tag{2}$$

Where:

- $n$: number of wards available the hospital structure;

- $AvLOS_j^{REF}$: Italian AvLOS in the specific hospital ward j;

- $AvLOS_{ALL}^{REF}$: Italian AvLOS considering all hospital wards;

- $d_j^{REF}$: number of patient discharges in Italy in the specific hospital ward j

- $d_{ALL}^{REF}$: total number of patient discharges in Italy;

- $AvLOS_j^{STR}$: AvLOS in the specific hospital ward j in the relevant structure;

- $d_j^{STR}$: number of patients discharged in the relevant structure in the specific hospital ward j;

- $d_{ALL}^{STR}$: total number of patients discharges in the relevant structure.

For instance, the two indices for the Lazio region are computed as follows:

$$CMI_{Lazio} = \frac{AvLOS_{cardiology}^{Italy}\left(\frac{d_{cardiology}^{Lazio}}{d_{All\_disciplines}^{Lazio}}\right) + AvLOS_{pneumology}^{Italy}\left(\frac{d_{pneumology}^{Lazio}}{d_{All\_disciplines}^{Lazio}}\right) + \cdots + AvLOS_{ICU}^{Italy}\left(\frac{d_{ICU}^{Lazio}}{d_{All\_disciplines}^{Lazio}}\right)}{AvLOS_{All\_disciplines}^{Lazio}}$$

$$PI_{Lazio} = \frac{AvLOS_{cardiology}^{Lazio}\left(\frac{d_{cardiology}^{Italy}}{d_{All\_disciplines}^{Italy}}\right) + AvLOS_{pneumology}^{Lazio}\left(\frac{d_{pneumology}^{Italy}}{d_{All\_disciplines}^{Italy}}\right) + \cdots + AvLOS_{ICU}^{Lazio}\left(\frac{d_{ICU}^{Italy}}{d_{All\_disciplines}^{Italy}}\right)}{AvLOS_{All\_disciplines}^{Lazio}}$$

A high CMI value (Eq 1) indicates a more complex and resource-intensive case load managing of clinical cases. In particular a region with a CMI higher than 1 tends to hospitalize patients in wards with a high average length of stay in comparison to the national benchmark. Conversely, a high PI (Eq 2) value is found when the hospital length of stay is longer than expected. In particular, a region with a PI higher than 1, assuming equal complexity, tends to hospitalize patients for longer periods, thus suggesting lower efficiency relative to the standard [21, 22, 32]. Considering the graphical representation, these indicators are analysed adopting a four-quadrant graph where the CMI is reported in the abscissa and the PI is reported in the ordinate compared to the national benchmark.

## Results

### Impact of the COVID-19 outbreak in the hospital bed management in Italy

Since the beginning of the contagion, the number of patients infected follows an exponential trend, with a slight change during the last week turning from an exponential to a linear trend (Fig 1). This tendency is evident considering, on the one hand, the total number cases, the number of deaths and healed patients, and, on the other, the number of patients hospitalized in particular in the ICUs. Trends describing hospitalizations are reported in Fig 1 considering both the country as a whole and Lombardia in particular, being the region with the highest number of cases in Italy. These trends highlight the tremendous amount of work carried out by the healthcare professionals to host the exceptional number of patients in the hospital. While all hospital wards are struggling with an exceptional workload due to the COVID-19 outbreak, the ICUs are particularly stressed given that the majority of the hospitals in the northern part of the country are saturating their capacity. Fig 2 summarizes the actual occupancy rate of the ICU beds to date (29th of March 2020). It is important to note that the occupancy rate is computed considering the number of beds available in each region in 2017 and the number of hospitalized patients affected by COVID-19 and does not take into account patients admitted with other pathologies [33]. As highlighted by the colour, different regions have already saturated the hospital beds ordinarily available in their ICUs (Lombardia, Piemonte, Val d'Aosta, Trento, Bolzano), with other regions worryingly approaching this threshold (Marche, Emilia Romagna, Toscana, Liguria). It is to be noted that, all of them are located in the northern part of Italy.

For the time being, it is difficult to analyse the local health system management responses to the COVID-19 crisis as different components have to be considered including measures of lockdown, population characteristics, resources available in terms of health professionals, equipment, territorial services, etc. However, the analysis carried out in the next sections is in our opinion an important reference point to further analyse how and to what extent the health system reacted to this pandemic period.

### Analysis of the Italian structural components

In this paragraph data collected in the years 2010 and 2017 is compared to capture differences in terms of beds availability, number of hospitalizations and total number of inpatient days. A

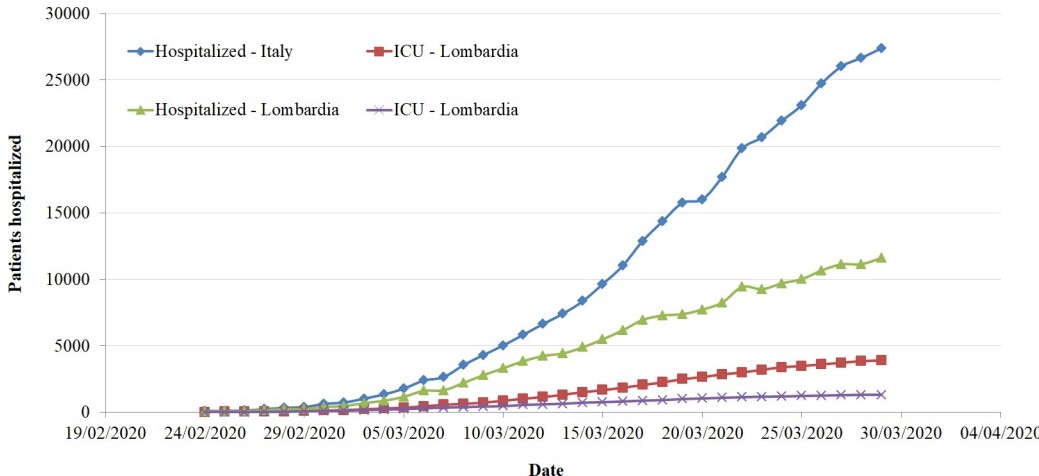

**Fig 1. Number of patients hospitalized in the Italian regions and in Lombardia by COVID-19 wards and Intensive Care Units [7].**

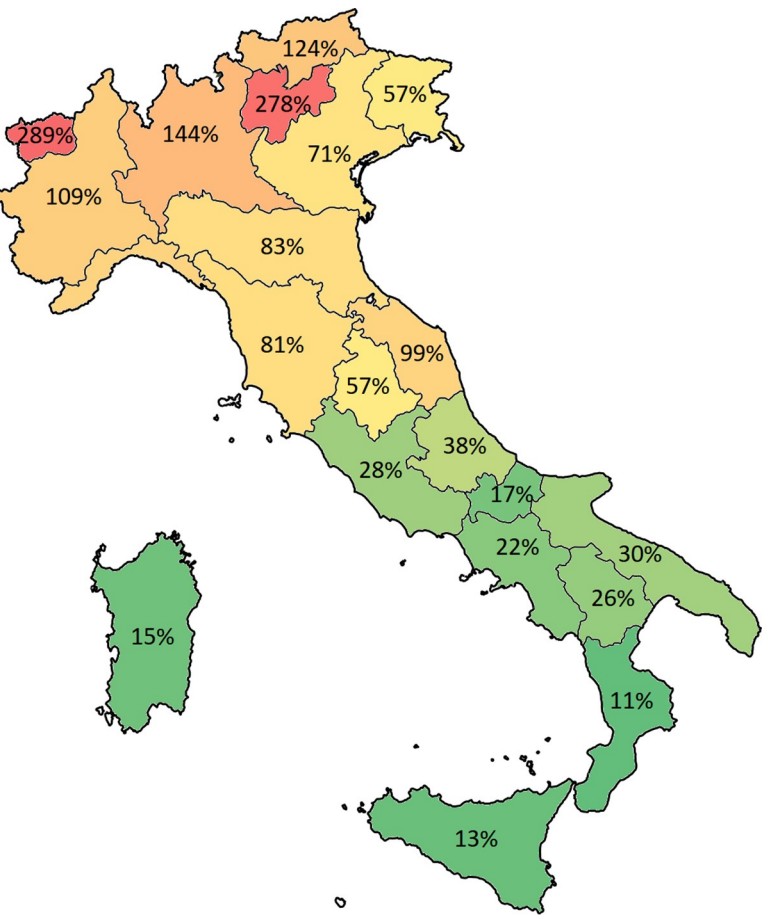

**Fig 2. Percent of routinely available beds in the Intensive Care Units occupied by COVID-19 patients by region.**
Colours indicate the level of bed occupancy rate spanning from red (high % of beds occupied) and green (low % of beds occupied).

highlighted in Table 1, an important reduction of the hospital beds is shown all over the Italian regions. This reduction matches with a reduction of both number of inpatients and total number of days spent in the hospital. Differences across the country indicate the highest level of reductions in the southern regions. Considering the northern regions heavily affected by the COVID-19, the reduction ranges from -5% in Bolzano and Val d'Aosta to -23% in Trento, while Lombardia cuts around the 7% of hospital beds.

A different picture is shown focusing on the ICUs. Table 2 highlights that the number of beds in these departments has not changed in the period considered, on the contrary there was an increase of around 6% in the number of beds all over the territory. Remarkably, differences across regions can be detected, even if only five regions have reduced the number of intensive care beds with the highest cut of 5% in Lazio, Liguria and Piemonte.

A focus on the reduction of the hospital beds in the different wards is reported in Table 3 highlighting those covering the 90% of the hospitalization complexity in terms of the average length of stay. In particular, oncology and ICUs have had an increase in the number of beds in Italy, while in other wards the reduction was even more than 20%, among which surgery, paediatrics and otolaryngology. This indicates that, among the wards that manage complex clinical cases, the availability of hospital beds in the ICUs was not affected by the funding cuts to public health. These crude differences may also be due to the tendency, widespread in the last

**Table 1. Number of hospital beds, hospitalization and length of stay in the years 2010 and 2017 highlighting the differences in each Italia region.**

| All hospital wards | 2010 | | | 2017 | | | % difference 2017–2010 | | |
|---|---|---|---|---|---|---|---|---|---|
| | Beds | Inpatients | Length of Stay | Beds | Inpatients | Length of Stay | Beds | Inpatients | Length of Stay |
| Abruzzo | 3688 | 140355 | 1044054 | 2966 | 118111 | 902896 | -20% | -16% | -14% |
| Basilicata | 1723 | 61958 | 464549 | 1654 | 57352 | 446263 | -4% | -7% | -4% |
| Calabria | 4231 | 170137 | 1156658 | 3168 | 129178 | 923884 | -25% | -24% | -20% |
| Campania | 11423 | 485809 | 3228706 | 9899 | 389661 | 2827162 | -13% | -20% | -12% |
| Emilia Romagna | 14278 | 526436 | 4231655 | 12808 | 493810 | 3753721 | -10% | -6% | -11% |
| Friuli Venezia Giulia | 3931 | 132046 | 1092524 | 3504 | 132105 | 1011899 | -11% | 0% | -7% |
| Lazio | 16472 | 600001 | 4905223 | 14213 | 489919 | 4016779 | -14% | -18% | -18% |
| Liguria | 5775 | 198075 | 1738658 | 4847 | 177825 | 1507474 | -16% | -10% | -13% |
| Lombardia | 30216 | 1085883 | 8644011 | 28047 | 952930 | 7956351 | -7% | -12% | -8% |
| Marche | 4802 | 177308 | 1349109 | 3969 | 145161 | 1151105 | -17% | -18% | -15% |
| Molise | 1335 | 48710 | 370750 | 1063 | 33597 | 263986 | -20% | -31% | -29% |
| Piemonte | 13175 | 443031 | 3984466 | 11604 | 381420 | 3394503 | -12% | -14% | -15% |
| P.A. Bolzano | 1660 | 66503 | 467972 | 1578 | 61154 | 445617 | -5% | -8% | -5% |
| P.A. Trento | 1756 | 50846 | 439402 | 1353 | 51594 | 400525 | -23% | 1% | -9% |
| Puglia | 11971 | 488098 | 3376215 | 9306 | 373040 | 2751312 | -22% | -24% | -19% |
| Sardegna | 4834 | 181711 | 1288677 | 4139 | 152247 | 1120080 | -14% | -16% | -13% |
| Sicilia | 11348 | 476575 | 3287241 | 10200 | 365911 | 3043748 | -10% | -23% | -7% |
| Toscana | 10719 | 431772 | 2960213 | 9139 | 372440 | 2603535 | -15% | -14% | -12% |
| Umbria | 2521 | 117888 | 777688 | 2565 | 102723 | 782340 | 2% | -13% | 1% |
| Valle d'Aosta | 401 | 14340 | 119400 | 379 | 13506 | 108196 | -5% | -6% | -9% |
| Veneto | 16236 | 527677 | 4735739 | 14638 | 500422 | 4244400 | -10% | -5% | -10% |
| **Total** | **172495** | **6425159** | **49662910** | **151039** | **5494106** | **43655776** | **-12%** | **-14%** | **-12%** |

years in many European countries, of adopting policies that strongly shift the organization and provision of health and social services from formal institutional facilities (e.g. hospitals) to home care [34]. Moreover, recently the provision of different scheduled procedures (e.g. diagnostic test, clinical examinations, treatments) are mainly provided on a day hospital basis, reducing the number of beds needed to treat the patients.

Other differences related to the availability of hospital beds refer to their distribution in the 21 regions and autonomous provinces. Fig 3 reports the number of beds per 100.000 inhabitants considering all disciplines (A) and focusing on the intensive care unit departments (B). The proportions vary across the country spanning from Molise that counts the highest values for both categories to Calabria that displays the lowest proportions. Summarizing, southern Italy (e.g. Abruzzo, Calabria, Campania, Puglia, Sicilia) suffers the lack of hospital beds and lays in the middle of the rank with 279 beds in total and 8 intensive care unit beds per 100.000 inhabitants, as shown in Fig 3.

## Hospital bed management

Table 4 reports the results obtained for each region highlighting the classification in the areas of hospital bed management as well as case-mix and performance indicators. Green cells highlight region where the indicators are above the relevant threshold (i.e. BOR between 75% and 85% and TOI between 1 and 3 days). Note that while BOR and TOI thresholds are from the literature [26–29], those for the AvLOS and BTO are computed on the basis of the national average given that reference values for these indicators strictly depend on the distribution of the admissions in the different hospital wards. For instance, in Italy, the average length of stay in

**Table 2. Number of hospital beds, hospitalization and length of stay in the years 2010 and 2017 in the intensive care units highlighting the differences in each Italian region.**

| Intensive care unit hospital ward | 2010 | | | 2017 | | | % difference 2017–2010 | | |
|---|---|---|---|---|---|---|---|---|---|
| | Beds | Inpatients | Length of Stay | Beds | Inpatients | Length of Stay | Beds | Inpatients | Length of Stay |
| Abruzzo | 91 | 1105 | 18566 | 92 | 1198 | 17723 | 1% | 8% | -5% |
| Basilicata | 41 | 698 | 10576 | 49 | 745 | 9782 | 20% | 7% | -8% |
| Calabria | 107 | 1666 | 24931 | 124 | 2741 | 29237 | 16% | 65% | 17% |
| Campania | 363 | 7239 | 94649 | 427 | 6502 | 91616 | 18% | -10% | -3% |
| Emilia Romagna | 372 | 3710 | 53817 | 360 | 4201 | 51943 | -3% | 13% | -3% |
| Friuli Venezia Giulia | 103 | 1743 | 14424 | 119 | 1936 | 16132 | 16% | 11% | 12% |
| Lazio | 510 | 4770 | 98347 | 486 | 4699 | 91136 | -5% | -1% | -7% |
| Liguria | 173 | 1843 | 28332 | 164 | 1491 | 25157 | -5% | -19% | -11% |
| Lombardia | 663 | 6775 | 92057 | 738 | 7641 | 91661 | 11% | 13% | 0% |
| Marche | 113 | 1608 | 18885 | 127 | 1538 | 19826 | 12% | -4% | 5% |
| Molise | 29 | 413 | 6930 | 35 | 496 | 6980 | 21% | 20% | 1% |
| Piemonte | 316 | 3428 | 50239 | 299 | 3468 | 44595 | -5% | 1% | -11% |
| P.A. Bolzano | 36 | 616 | 7205 | 37 | 612 | 4592 | 3% | -1% | -36% |
| P.A. Trento | 20 | 272 | 2673 | 31 | 369 | 3628 | 55% | 36% | 36% |
| Puglia | 197 | 3891 | 53895 | 231 | 3884 | 56404 | 17% | 0% | 5% |
| Sardegna | 107 | 1614 | 25445 | 120 | 1695 | 22436 | 12% | 5% | -12% |
| Sicilia | 336 | 5240 | 73888 | 323 | 5118 | 76188 | -4% | -2% | 3% |
| Toscana | 322 | 3415 | 49959 | 373 | 3911 | 56092 | 16% | 15% | 12% |
| Umbria | 61 | 683 | 8615 | 69 | 730 | 8914 | 13% | 7% | 3% |
| Valle d'Aosta | 8 | 118 | 911 | 10 | 118 | 1273 | 25% | 0% | 40% |
| Veneto | 448 | 4851 | 68142 | 468 | 4777 | 64813 | 4% | -2% | -5% |
| **Total** | **4416** | **55698** | **802486** | **4682** | **57870** | **790128** | **6%** | **4%** | **-2%** |

the otolaryngology ward is around 4 days, in the cardiology ward is around 6 days, whereas in the ICU is around 14 days. This difference also influences the bed turn over that indicates the number of patients per bed hospitalized during the year.

Considering the hospital bed management, all regions fall within the threshold values both for the BOR and the TOI, with the exception of Molise, Basilicata and Sardegna. Regional differences can be captured considering both AvLOS and BTO. The first one indicates regions (i.e. Lombardia and Liguria) that tend to extend the patient's hospital stays when compared to the national benchmarking value, while BTO captures regions (i.e. Lombardia and Molise) with a low number of patients as a percentage of available beds. The difference in the AvLOS across the country can determine the level of performance of each region on the basis of the complexity of cases. On the basis of these analyses, regions are classified in the following four macro-clusters, that mainly represent the classification of the complex and performance indicators (Fig 4):

1. Six regions (Abruzzo, Emilia Romagna, Trento, Toscana, Umbria e Basilicata) reported positive results in the management of hospital beds. These regions report high performances in the treatment of complex cases, mainly due to short hospitalization with respect to the national average length of stay.

2. The second group of regions (Calabria, Campania, Bolzano e Puglia) has similar results compared to the above cluster, with hospitalizations efficiently managed with high performances. Differently from the above-mentioned group, these regions tend to manage cases that are less complex.

**Table 3. Number of beds in the years 2010 and 2017 highlighting the differences in each hospital ward.**

| Beds available in each hospital ward | 2010 | 2017 | % Difference 2010–2017 | ICM reference value |
|---|---|---|---|---|
| General medicine | 30118 | 26053 | -13% | 0,20 |
| General surgery | 19854 | 15889 | -20% | 0,11 |
| Orthopaedics and traumatology | 14314 | 12063 | -16% | 0,08 |
| Obstetrics and gynaecology | 13000 | 10934 | -16% | 0,07 |
| Function recovery and rehabilitation | 10049 | 9708 | -3% | 0,06 |
| Cardiology | 6759 | 6687 | -1% | 0,05 |
| Long-term patients | 5135 | 4067 | -21% | 0,03 |
| Neurology | 5222 | 4944 | -5% | 0,03 |
| Paediatrics | 5746 | 4578 | -20% | 0,03 |
| Geriatrics | 4061 | 3550 | -13% | 0,03 |
| Urology | 5139 | 4501 | -12% | 0,03 |
| Psychiatry | 4172 | 4032 | -3% | 0,03 |
| Pneumology | 3777 | 3186 | -16% | 0,02 |
| Oncology | 2703 | 2759 | 2% | 0,02 |
| Infectious and tropical diseases | 3154 | 2816 | -11% | 0,02 |
| Neurosurgery | 2733 | 2454 | -10% | 0,02 |
| Intensive care | 4416 | 4682 | 6% | 0,02 |
| Otolaryngology | 3408 | 2549 | -25% | 0,02 |
| Neonatology | 1944 | 1913 | -2% | 0,01 |
| Nephrology | 2049 | 1913 | -7% | 0,01 |
| Neuro-rehabilitation | 1981 | 1855 | -6% | 0,01 |
| Gastroenterology | 1716 | 1616 | -6% | 0,01 |

3. Six regions (Friuli Venezia Giulia, Piemonte, Valle D'Aosta, Veneto, Liguria e Lombardia) manage the hospital beds with a high turnover and beds that remain scarcely empty during the year. This is mainly due to a high length of stay that results in a low level of performance compared to the national level, even if these regions mainly manage complex cases.

4. The last group is composed by four regions (Lazio, Marche, Sicilia e Sardegna). Even if they have positive values in the hospital bed turnover and turnover interval, the performance is

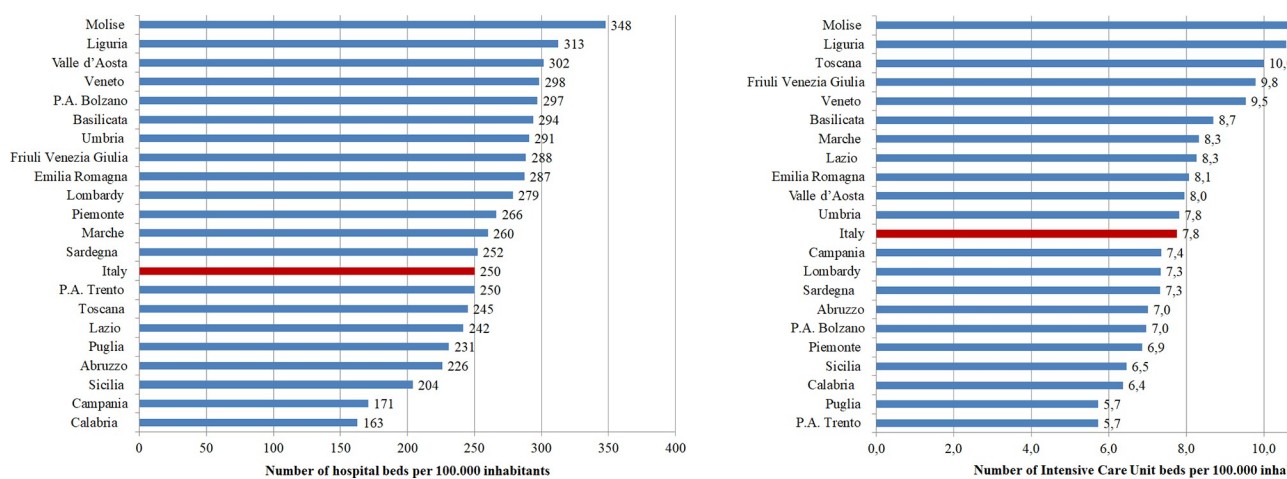

**Fig 3. Number of total and intensive care unit beds per 100000 inhabitants in the different Italian regions.**

**Table 4. Cross-regional comparison of the results of bed management as well as complex and performance analysis.**

| All hospital wards | TOI | BOR | AvLOS | BTO | CMI | PI |
|---|---|---|---|---|---|---|
| Abruzzo | 1,52 | 83% | 7,64 | 39,82 | 1,02 | 0,97 |
| Basilicata | 2,75 | 74% | 7,78 | 34,67 | 1,03 | 0,97 |
| Calabria | 1,80 | 80% | 7,15 | 40,78 | 0,95 | 0,98 |
| Campania | 2,02 | 78% | 7,26 | 39,36 | 0,94 | 0,99 |
| Emilia Romagna | 1,87 | 80% | 7,60 | 38,55 | 1,02 | 0,96 |
| Friuli Venezia Giulia | 2,02 | 79% | 7,66 | 37,70 | 1,01 | 1,00 |
| Lazio | 2,39 | 77% | 8,20 | 34,47 | 0,96 | 1,08 |
| Liguria | 1,47 | 85% | 8,48 | 36,69 | 1,08 | 1,01 |
| Lombardia | 2,39 | 78% | 8,35 | 33,98 | 1,03 | 1,04 |
| Marche | 2,05 | 79% | 7,93 | 36,57 | 0,96 | 1,06 |
| Molise | 3,69 | 68% | 7,86 | 31,61 | 1,03 | 1,02 |
| Piemonte | 2,20 | 80% | 8,90 | 32,87 | 1,03 | 1,10 |
| P.A. Bolzano | 2,13 | 77% | 7,29 | 38,75 | 0,99 | 0,93 |
| P.A. Trento | 1,81 | 81% | 7,76 | 38,13 | 1,01 | 0,98 |
| Puglia | 1,73 | 81% | 7,38 | 40,09 | 0,95 | 0,99 |
| Sardegna | 2,57 | 74% | 7,36 | 36,78 | 0,97 | 1,05 |
| Sicilia | 1,86 | 82% | 8,32 | 35,87 | 0,97 | 1,08 |
| Toscana | 1,97 | 78% | 6,99 | 40,75 | 1,00 | 0,91 |
| Umbria | 1,50 | 84% | 7,62 | 40,05 | 1,04 | 0,94 |
| Valle d'Aosta | 2,23 | 78% | 8,01 | 35,64 | 1,02 | 1,08 |
| Veneto | 2,20 | 79% | 8,48 | 34,19 | 1,03 | 1,04 |
| **Total** | **2,09** | **79%** | **7,95** | **36,38** | | |

lower than the national reference value also considering that these regions tend to manage fewer complex cases.

In this classification Molise represents an interesting outlier. It is the only region with both BOR and TOI outside the efficiency thresholds. This result may be associated with the high number of hospital beds per 100.000 inhabitants, that is the highest number across the 21

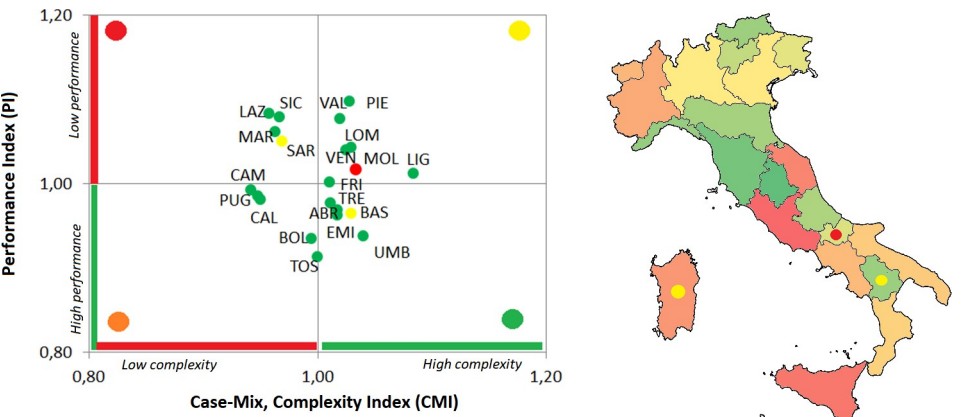

**Fig 4. Overall classification of each Italian region considering both the hospital bed management indicators and the complexity and performance indicators.** In both figures the colour of the marker captures the hospital bed management classification. In the Italian map, regions are coloured on the basis of the ratio between the CMI and PI.

**Table 5. Cross-regional comparison of the results of bed management as well as complex and performance analysis of intensive care units.**

| Intensive Care Unit | TOI | BOR | AvLOS | BTO | CMI | CPI |
|---|---|---|---|---|---|---|
| Abruzzo | 13,24 | 53% | 14,79 | 13,02 | 0,97 | 1,14 |
| Basilicata | 10,88 | 55% | 13,13 | 15,20 | 1,25 | 1,02 |
| Calabria | 5,85 | 65% | 10,67 | 22,10 | 2,05 | 0,81 |
| Campania | 9,88 | 59% | 14,09 | 15,23 | 1,58 | 1,04 |
| Emilia Romagna | 18,91 | 40% | 12,36 | 11,67 | 0,81 | 0,91 |
| Friuli Venezia Giulia | 14,10 | 37% | 8,33 | 16,27 | 1,39 | 0,62 |
| Lazio | 18,36 | 51% | 19,39 | 9,67 | 0,91 | 1,42 |
| Liguria | 23,27 | 42% | 16,87 | 9,09 | 0,80 | 1,24 |
| Lombardia | 23,26 | 34% | 12,00 | 10,35 | 0,76 | 0,88 |
| Marche | 17,25 | 43% | 12,89 | 12,11 | 1,01 | 0,95 |
| Molise | 11,68 | 55% | 14,07 | 14,17 | 1,41 | 1,14 |
| Piemonte | 18,61 | 41% | 12,86 | 11,60 | 0,86 | 0,95 |
| P.A. Bolzano | 14,56 | 34% | 7,50 | 16,54 | 0,96 | 0,60 |
| P.A. Trento | 20,83 | 32% | 9,83 | 11,90 | 0,68 | 0,75 |
| Puglia | 7,19 | 67% | 14,52 | 16,81 | 0,98 | 1,08 |
| Sardegna | 12,60 | 51% | 13,24 | 14,13 | 1,05 | 0,98 |
| Sicilia | 8,15 | 65% | 14,89 | 15,85 | 1,33 | 1,09 |
| Toscana | 20,47 | 41% | 14,34 | 10,49 | 1,00 | 1,05 |
| Umbria | 22,29 | 35% | 12,21 | 10,58 | 0,67 | 0,91 |
| Valle d'Aosta | 20,14 | 35% | 10,79 | 11,80 | 0,85 | 0,87 |
| Veneto | 22,19 | 38% | 13,57 | 10,21 | 0,90 | 1,00 |
| **Total** | **15,88** | **46%** | **13,65** | **12,36** | | |

Italian regions. These features contribute also to the performance of the regional hospitals even if this region tends to manage complex cases.

To further detail this analysis, intensive care unit beds and hospitalization flow were analysed. Table 5 reports the results of the hospital bed management, complexity and performance indicators. Also in this case, green cells identify regions where the CMI and/or the PI values are above the benchmarking level (i.e. CMI higher than 1, PI lower than 1). In this analysis the thresholds for the four hospital bed management indicators are defined on the basis of the national average values, given that no reference values specifically focused on the ICU bed management are available in the literature. The macro-clusters detected on the basis of both analyses are summarized in the following (Fig 5):

1. In three regions (Calabria, Sardegna e Sicilia) all indicators report positive values, when considering the national average ones. They manage complex cases with high performance results. The quick turnover and a high percentage of bed occupancy make these regions efficient also in the management of ICU beds.

2. A consistent number of regions (Basilicata, Campania, Molise, Sicilia, Abruzzo, Puglia e Lazio) belong to the second cluster. As reported for the previous group, these regions have a high bed turnover and the number of days between two hospitalization are relatively low. Differently from the above-mentioned group, these results are mainly due to the high number of days that patients spend in the ICU. For this reason, even if these regions tend to cope with complex cases and pathologies, their performance is lower than the national one.

3. In three regions (Liguria, Toscana e Veneto) hospital beds are not efficiently managed with a low performance in the management of not so complex cases. This is mainly due to the low complexity of cases with a length of stay higher than the national average value.

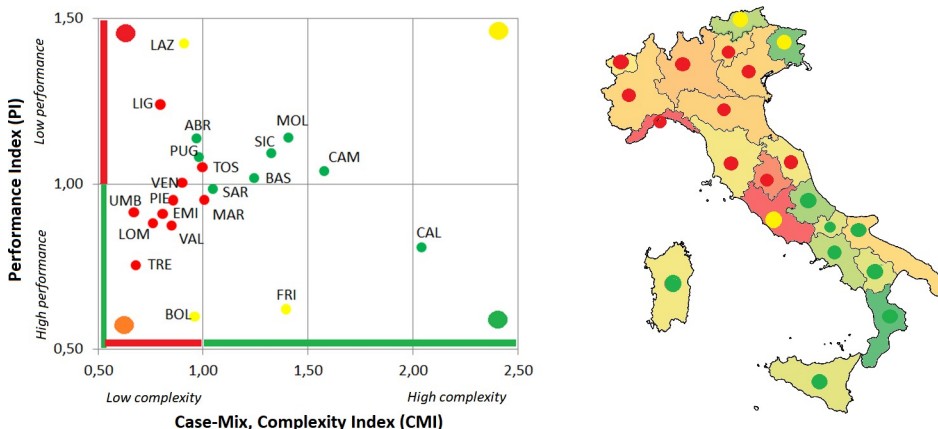

**Fig 5. Overall classification of each Italian region considering both the hospital bed management indicators and the complexity and performance indicators of intensive care units.** In both figures the colour of the marker captures the hospital bed management classification. In the Italian map, regions are coloured on the basis of the ratio between the CMI and PI.

4. In the last group of regions (Emilia Romagna, Lombardia, Piemonte, Trento, Umbria, Val D'Aosta, Bolzano) the majority of the hospitalization cases are not efficiently managed with a high value of length of stay for not complex cases.

In this classification Marche represents an outlier in the management of hospital beds. Differently from the last reported cluster, this region manages complex cases with a relative high performance.

## Discussion and conclusions

COVID-19 is one of the most serious pandemics of the past 100 years, its rapid global spread is overwhelming hospitals and local communities in Italy and worldwide. One of the main challenges is to rapidly and efficiently assign and reallocate appropriate resources, such as medical professionals, equipment, hospital beds to face overload and saturation. For these reasons, the analysis of hospital capacity and the efficiency in the management of its structures before the emergency outbreak provides an important reference point to further explore how the management of the emergency has been carried out. Moreover, on this basis lessons learned outlining structural bottle necks and/or facilitators could give indications on the best way to achieve hospital disaster preparedness in case of a COVID-19 second wave or other possible pandemics [35].

On the basis of the recent financial cuts of public health facilities, this paper analyses the efficiency in the management of hospital beds across Italian regions. The results of the overall analysis firstly show that even if an important reduction of hospital beds affected the majority of the hospital wards, an opposite trend has been detected for the ICUs that are particularly stressed in a respiratory pandemic crisis. This is particularly evident considering the regions that are most effected by the virus where the number of ICU beds has increased in recent years (i.e. between 2010 and 2017): Lombardia +11%, Marche +12%, Piemonte -5%, Trento +55%, Bolzano +3%, Veneto +4%.

Hospital beds are generally efficiently managed all over the country with some exceptions in the south of Italy that show a slow turnover and/or a low bed occupancy rate. The Italian northern regions show instead that cases are handled with rapid shifts and without leaving

beds empty during the year. Moreover, generally northern regions mainly deal with complex cases, albeit with a performance below the national average. On the contrary, different results are displayed analysing the management of beds in the intensive care units, where the 75% occupancy rate threshold is not reached in all regions. In this sense, the northern regions exhibit a high performance, even if related to the management of less complex cases that can be related to a large patient mobility towards hospitals located in the northern regions especially for elective treatments.

Moreover, there was no substantial reduction of beds in ICUs, if compared to relevant financial cuts in other wards. This trend may be explained by the attempt to reorganizing the national public health reducing hospital costs at the same time. On the one hand, the provision of health services is increasingly shifting from formal institutional facilities (e.g. hospitals) to home care and, on the other hand, different scheduled procedures are mainly provided on a day hospital basis reducing the number of beds needed to treat the patients. The majority of the Italian regions and in particular those in the northern part of the country can rely on an appropriate number of beds that generally do not saturate in normal periods. The availability of hospital beds as well as the efficiency in their management confirmed in this study allow hospitals to treat patients without the risk of having an overabundance of patients and a scarcity of beds. In fact, the analysis reported in this paper showed that, in normal situations, the management of hospital has no critical levels. This is particularly evident considering the ICU wards where the BOR is lower than 67% in all regions and even lower in the regions that have saturated the ICU beds during the epidemics spread: Lombardia 34%, Marche 43%, Piemonte 41%, Trento 32%, Bolzano 34%, Veneto 38%. During pandemic or other catastrophic periods, the hospital management paradigms need to be changed [36] making it necessary to balance the relationship between hospital and territory as well as to determine the appropriate allocation of inpatient resources [37]. The availability of hospitalization data as well as the continuous change in terms of bed availability across the Italian regions make it difficult to apply this methodology in this pandemic period yet. However, an ex post analysis that relies in updated and robust data should be applied in the future to provide indications on the hospital-territory relationship in response citizens' safety and wellbeing.

## Author Contributions

**Conceptualization:** Fabrizio Clemente, Daniela Luzi.

**Data curation:** Fabrizio Pecoraro.

**Formal analysis:** Fabrizio Pecoraro.

**Methodology:** Fabrizio Pecoraro.

**Supervision:** Fabrizio Pecoraro, Fabrizio Clemente, Daniela Luzi.

**Validation:** Fabrizio Clemente, Daniela Luzi.

**Writing – original draft:** Fabrizio Pecoraro.

**Writing – review & editing:** Fabrizio Pecoraro, Fabrizio Clemente, Daniela Luzi.

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
