## [Decision Letter · Decision Letter 0]

9 Jun 2020

PONE-D-20-09436

The efficiency in the ordinary hospital bed management in Italy: an in-depth analysis of intensive care unit in the areas affected by COVID-19 before the outbreak

PLOS ONE

Dear Dr. Pecoraro,

Thank you for submitting your manuscript to PLOS ONE. After careful consideration, we feel that it has merit but does not fully meet PLOS ONE’s publication criteria as it currently stands. Therefore, we invite you to submit a revised version of the manuscript that addresses the points raised during the review process.

We look forward to receiving your revised manuscript.

Kind regards,

Maria Michela Gianino

Academic Editor

PLOS ONE

2. Please refer to the specific statistical analyses performed in your Methods section; and illustrate the results of such analysis in more detail in the Results (for more information, please see https://journals.plos.org/plosone/s/submission-guidelines.#loc-statistical-reporting). Moreover, please provide more background on the choice of using the year 2010 as a control.

4. We note that Figure 2 in your submission contain [map/satellite] images which may be copyrighted. All PLOS content is published under the Creative Commons Attribution License (CC BY 4.0), which means that the manuscript, images, and Supporting Information files will be freely available online, and any third party is permitted to access, download, copy, distribute, and use these materials in any way, even commercially, with proper attribution. For these reasons, we cannot publish previously copyrighted maps or satellite images created using proprietary data, such as Google software (Google Maps, Street View, and Earth). For more information, see our copyright guidelines: http://journals.plos.org/plosone/s/licenses-and-copyright.

Reviewers' comments:

Reviewer's Responses to Questions

**Comments to the Author**

1. Is the manuscript technically sound, and do the data support the conclusions?

Reviewer #1: Partly

Reviewer #2: Partly

2. Has the statistical analysis been performed appropriately and rigorously? 

Reviewer #1: I Don't Know

Reviewer #2: No

3. Have the authors made all data underlying the findings in their manuscript fully available?

Reviewer #1: Yes

Reviewer #2: Yes

4. Is the manuscript presented in an intelligible fashion and written in standard English?

Reviewer #1: No

Reviewer #2: Yes

5. Review Comments to the Author

Reviewer #1: “The efficiency in the ordinary hospital bed management in Italy: an in-depth analysis of intensive care unit in the areas affected by COVID-19 before the outbreak” presents an analysis of bed availability in Italy prior to the epidemic. It concludes that bed supply was adequate and utilized efficiently at this time. Another important question is whether bed supply was adequate to match demand due to the pandemic. Obviously, the demand increased by such large amounts that it was not adequate especially in northern regions. It would be informative for the authors to extend their analysis to address this question. Also, I found the paper confusing and overly detailed in parts. Some specific comments/questions include:

• Provide examples of the calculation of CMI and PI, perhaps in an appendix. Provide a clearer explanation of Figure 2. I assume the numbers within each region represent N of Covid19 pts/N of ICU beds. If so the title could be revised to read, “Percent of routinely available beds occupied by COVID-19 patients by region. Colours indicate ...”

• What is the definition of a specialty (j)? It would be better to do a case-mix adjustment using the diagnosis/DRG of individual patients to provide more sensitivity to differences among regions. Even better would be to add an admission severity of illness score to the diagnosis. If data aren’t available to do this, this limitation in the case-mix adjustment method should be stated.

• Is there any audit or other mechanism to ensure reliability of source data? If not, is there any evidence that there may be differences in diagnostic coding or other data elements among regions?

• Table 1 columns don’t line up with headings. And are the total percentages at the bottom simple or weighted averages?

• On Lines 211-212 it states, “indicators are above the relevant threshold”. The authors should be explicit about the threshold. They should also be consistent. One line 242 they state, “Also in this case, green cells identify efficient regions.” Is this what is meant by “above the relevant threshold”?

• On Line 212 it states “BOR and TOI thresholds are captured in the literature”. What literature are you citing?

• On Line 135 the term “efficacy” is used. I think the authors mean “efficiency” here? If not, “efficacy” should be defined and how it’s measured made explicit.

• Putting “High” and “Low” on Figure 4 and Figure 5 would help the reader know which is the desirable direction.

• There are several grammatical errors, for example the disagreement of the singular nouns and plural verbs in the following:

o 57 highlighted that a reduction in the number of beds have affected many hospitals over the territory.

o 64 was published by Rhodes et al in 2012 [7]. Also in this case the number of beds located in Italy are

o 239 regional hospitals even if this region tend to manage complex cases.

Reviewer #2: The topic is relevant in terms of the present global pandemic situation. However, the aim of this research is not adequate in terms of an international journal perspective.

Data presented using summary statistics is useful, but inclusion of bar charts and graphs whilst comparing the data could be more relevant.

Line 68 ......ending with 'Government has a weak strategic leadership' - Ref required

In general, discussion is too short; there are more points/results to be discussed and more references needs to be included rather than giving vague statements.

Discussion and conclusion could be separated

6. PLOS authors have the option to publish the peer review history of their article (what does this mean?). If published, this will include your full peer review and any attached files.

Reviewer #1: No

Reviewer #2: Yes: Dr. A Peter

---

## [Author Response · Author response to Decision Letter 0]

20 Jul 2020

1. Please ensure that your manuscript meets PLOS ONE's style requirements, including those for file naming. The PLOS ONE style templates can be found at: https://journals.plos.org/plosone/s/file?id=wjVg/PLOSOne_formatting_sample_main_body.pdf and https://journals.plos.org/plosone/s/file?id=ba62/PLOSOne_formatting_sample_title_authors_affiliations.pdf

2. Please refer to the specific statistical analyses performed in your Methods section; and illustrate the results of such analysis in more detail in the Results (for more information, please see https://journals.plos.org/plosone/s/submission-guidelines.#loc-statistical-reporting). Moreover, please provide more background on the choice of using the year 2010 as a control. 

Author’s response: The quantitative analysis proposed in this paper is based on a computation of indices with the aim of assessing the efficiency of hospital bed management and the complexity of cases treated. A proper statistical analysis (regression, analysis of variance, etc.) is not applied, while the trend of those indices between available data (year 2010 and 2017) has been considered and discussed. Considering the choice of year 2010 as a control, we compared 2017 with 2010 as they are the last and the first data available from Italian Ministry of Health. 

3. We suggest you thoroughly copyedit your manuscript for language usage, spelling, and grammar. If you do not know anyone who can help you do this, you may wish to consider employing a professional scientific editing service. Whilst you may use any professional scientific editing service of your choice, PLOS has partnered with both American Journal Experts (AJE) and Editage to provide discounted services to PLOS authors. Both organizations have experience helping authors meet PLOS guidelines and can provide language editing, translation, manuscript formatting, and figure formatting to ensure your manuscript meets our submission guidelines. To take advantage of our partnership with AJE, visit the AJE website (http://learn.aje.com/plos/) for a 15% discount off AJE services. To take advantage of our partnership with Editage, visit the Editage website (www.editage.com) and enter referral code PLOSEDIT for a 15% discount off Editage services. If the PLOS editorial team finds any language issues in text that either AJE or Editage has edited, the service provider will re-edit the text for free.

Author’s response: All the authors carefully read and improved the manuscript which was also revised by a professional mother tongue translator appointed by IRPPS. 

4. We note that Figure 2 in your submission contain [map/satellite] images which may be copyrighted. 

Author’s response: The map reported in Table 2 has been changed adopting a figure downloaded from the Wikipedia website and painted by authors. The file, as stated by Wikipedia, is in the public domain and the author should “grant anyone the right to use this work for any purpose, without any conditions, unless such conditions are required by law”. https://it.wikipedia.org/wiki/File:Italy_map_with_regions.svg

Reviewers' comments. 

Reviewer #1: 

“The efficiency in the ordinary hospital bed management in Italy: an in-depth analysis of intensive care unit in the areas affected by COVID-19 before the outbreak” presents an analysis of bed availability in Italy prior to the epidemic. It concludes that bed supply was adequate and utilized efficiently at this time. Another important question is whether bed supply was adequate to match demand due to the pandemic.

Author’s response: We improved the discussion and conclusions paragraph to underline this aspect. Starting from the results of the hospital bed management analysis it is clear that the majority of the Italian regions can rely on an appropriate number of beds that generally do not saturate in normal periods. This is true also considering the ICU wards. Of course, during pandemic or other catastrophic periods, the hospital management paradigms need to be changed making it necessary to balance the relationship between hospital and territory as well as to determine the appropriate allocation of inpatient resources. However, to apply this methodology in this pandemic period it is necessary to perform an ex post analysis that relies in updated and robust data. In our view the analysis we performed has to be taken into account when considering local health management responses to pandemics.

Obviously, the demand increased by such large amounts that it was not adequate especially in northern regions. It would be informative for the authors to extend their analysis to address this question. Also, I found the paper confusing and overly detailed in parts. Some specific comments/questions include:

Author’s response: The text and style of the paper was improved. 

• Provide examples of the calculation of CMI and PI, perhaps in an appendix. 

Author’s response: Now an example of how the two indices are computed for a given region (i.e. Lazio) is reported in the M&M section. We also included additional references to better explain these indices. 

• Provide a clearer explanation of Figure 2. I assume the numbers within each region represent N of Covid19 pts/N of ICU beds. If so the title could be revised to read, “Percent of routinely available beds occupied by COVID-19 patients by region. Colours indicate ...”

Author’s response: Caption of figure 2 has been changed taking into account the reviewer’s suggestion. The actual version is: Figure 2. Percent of routinely available beds in the Intensive Care Units occupied by COVID-19 patients by region. Colours indicate the level of bed occupancy rate spanning from red (high % of beds occupied) and green (low % of beds occupied).

• What is the definition of a specialty (j)? It would be better to do a case-mix adjustment using the diagnosis/DRG of individual patients to provide more sensitivity to differences among regions. Even better would be to add an admission severity of illness score to the diagnosis. If data aren’t available to do this, this limitation in the case-mix adjustment method should be stated.

Author’s response: “Specialty” has been replaced with “ward” as the CMI and PI indices are computed taking into account the availability of beds and the hospital discharges in each hospital ward. Note that in the Italian hospital system each ward correspond to a clinical speciality. Considering data availability, information is extracted from a national data bank that exposes data aggregated by hospital facility and hospital ward. Moreover, the CMI and PI indices are based on hospital wards where the patients are hospitalized instead of DRGs. This part of the M&M has been updated to clarify the type of data source available and the methodology adopted to compute the indices.

• Is there any audit or other mechanism to ensure reliability of source data? If not, is there any evidence that there may be differences in diagnostic coding or other data elements among regions?

Author’s response: We elaborated data gathered from two data sources. The first one regards the hospital bed management and considers the Discharge Report Form sent by each hospital to the relevant region and then to the Italian Ministry of Health. This data are gathered and exposed by the Ministry of Health in its website and this information flow is part of the NSIS New Health Information System (NSIS) developed and managed by law by the Ministry of Health. In this work, as these official data are aggregated for clinical wards, we did not consider the diagnostic code (ICD or DRG), but we took into account the hospital ward where the patient has been admitted. The second source of information provides daily data on the virus diffusion during the COVID-19 outbreak. It exposes official and continuously updated information produced by the Italian Civil Protection Department and adopted by the Ministry of Health for its periodic bulletin (http://www.salute.gov.it/portale/nuovocoronavirus/archivioNotizieNuovoCoronavirus.jsp).

• Table 1 columns don’t line up with headings. And are the total percentages at the bottom simple or weighted averages?

Author’s response: Now the heading of Table 1 has been updated. The last raw represents values of Italy, thus it is not a simple average, but it highlights weighted differences in the whole country.

• On Lines 211-212 it states, “indicators are above the relevant threshold”. The authors should be explicit about the threshold. They should also be consistent. 

Author’s response: We added thresholds for the BOR and TOI indices as well as the relevant references. 

• One line 242 they state, “Also in this case, green cells identify efficient regions.” Is this what is meant by “above the relevant threshold”?

Author’s response: We added thresholds for the CMI and PI indices as well as the relevant references. 

• On Line 212 it states “BOR and TOI thresholds are captured in the literature”. What literature are you citing?

Author’s response: We added reference to specify thresholds of indices. 

• On Line 135 the term “efficacy” is used. I think the authors mean “efficiency” here? If not, “efficacy” should be defined and how it’s measured made explicit.

Author’s response: The correct term is efficiency. Authors apologise for these errors.

• Putting “High” and “Low” on Figure 4 and Figure 5 would help the reader know which is the desirable direction. 

Author’s response: Figure 4 and figure 5 have been updated on the basis of the reviewer’s suggestion 

• There are several grammatical errors, for example the disagreement of the singular nouns and plural verbs in the following:

o 57 highlighted that a reduction in the number of beds have affected many hospitals over the territory. 

o 64 was published by Rhodes et al in 2012 [7]. Also in this case the number of beds located in Italy are

o 239 regional hospitals even if this region tend to manage complex cases

Author’s response: The text has been further revised and improved by the authors and revised by a professional mother tongue translator.

Reviewer #2. 

The topic is relevant in terms of the present global pandemic situation. However, the aim of this research is not adequate in terms of an international journal perspective. 

Author’s response: This paper is focused on the Italian regions considering the impact of the virus on the Italian population in the first phase of the spread in Europe. Moreover, the results of this analysis may be adopted and extended to other countries in Europe to capture the importance of the availability of hospital resources to limit the spread of the virus and to care for patients with COVID-19. The aim as well as the hypothesis of the paper have been clarified in the updated version of the abstract and introduction. Moreover, the findings of the research have been further explored and argued in the discussion paragraph.

Data presented using summary statistics is useful, but inclusion of bar charts and graphs whilst comparing the data could be more relevant.

Author’s response: The paper has already a five diagrams and five figures. Moreover, we charted all the data reported in the tables to highlight results of the hospital bed management analysis. However, if the reviewer and the editor suggest additional data to be charted or graphed we are willing to graph them. 

• Line 68 ......ending with 'Government has a weak strategic leadership' - Ref required. 

Author’s response: Two references explaining this sentence have been added in the manuscript. They are: 

• Lo Scalzo A, Donatini A, Orzella L, Cicchetti A, Profi li S, Maresso A. Italy: Health system review. Health Systems in Transition. 2009; 11(6)1-216. 

• Armocida B, Formenti B, Ussai S, Palestra F, Missoni E. The Italian health system and the COVID-19 challenge. The Lancet Public Health. 2020; 5(5), e253.

• In general, discussion is too short; there are more points/results to be discussed and more references needs to be included rather than giving vague statements.

Author’s response: Discussion and conclusions paragraph has been improved. 

• Discussion and conclusion could be separated

Author’s response: We prefer to maintain discussion and conclusions in the same paragraph to give more continuity to the presentation of findings.

---

## [Decision Letter · Decision Letter 1]

3 Sep 2020

The efficiency in the ordinary hospital bed management in Italy: an in-depth analysis of intensive care unit in the areas affected by COVID-19 before the outbreak

PONE-D-20-09436R1

Dear Dr. Pecoraro,

We’re pleased to inform you that your manuscript has been judged scientifically suitable for publication and will be formally accepted for publication once it meets all outstanding technical requirements.

The decision depended on the following considerations:

- The authors have substantially clarified the presentation of their paper and the have provided a detailed response to each reviewer/editorial point raised

-  Reviewer 1 proposed a major revison by proposing “ It would be better to do a case-mix adjustment using the diagnosis/DRG of individual patients to provide more sensitivity to differences among regions. Even better would be to add an admission severity of illness score to the diagnosis. If data aren’t available to do this, this limitation in the case-mix adjustment method should be stated.”<o:p></o:p>

The reviewer therefore suggested using DRGs, but if the data were not available, the reviewer suggests to insert this lack as a limit. The reviewer did not make this adjustment binding.

- There are reviewers' comments on the validity of some indicators: “In addition, the measure of performance, bed occupancy rate, can be affected by other factors such as substitutability (e.g. an obstetrics patient can’t be put in infectious disease unit and vice-versa), regional differences in demand patterns (e.g. resort areas have more fluctuation over the year), and the size of a ward.” However, these indicators have not undergone a major revision. <o:p></o:p>

 <o:p></o:p>

-          While agreeing that the case-mix can represent a limitation, the article addresses a health policy problem that is particularly debated in Italy. As a consequence of the Ministerial Decree / 70, an intervention was carried out to reduce the number of beds in hospitals throughout Italy. With the COVID emergency, hospitalizations have increased and the difficulty of hospitals has been attributed to the DM / 70. The article assesses whether this widespread opinion is acceptable.<o:p></o:p>

<o:p></o:p>

 <o:p></o:p>

<o:p></o:p>

Kind regards,

Carmen Melatti

Staff Editor

PLOS ONE

on behalf of 

Maria Michela Gianino

Academic Editor

PLOS ONE

Additional Editor Comments (optional):

Reviewers' comments:

Reviewer's Responses to Questions

**Comments to the Author**

1. If the authors have adequately addressed your comments raised in a previous round of review and you feel that this manuscript is now acceptable for publication, you may indicate that here to bypass the “Comments to the Author” section, enter your conflict of interest statement in the “Confidential to Editor” section, and submit your "Accept" recommendation.

Reviewer #1: (No Response) 

Reviewer #2: All comments have been addressed

2. Is the manuscript technically sound, and do the data support the conclusions?

Reviewer #1: No

Reviewer #2: Yes

3. Has the statistical analysis been performed appropriately and rigorously? 

Reviewer #1: No

Reviewer #2: Yes

4. Have the authors made all data underlying the findings in their manuscript fully available?

Reviewer #1: Yes

Reviewer #2: Yes

5. Is the manuscript presented in an intelligible fashion and written in standard English?

Reviewer #1: Yes

Reviewer #2: Yes

6. Review Comments to the Author

Reviewer #1: The authors have substantially clarified the presentation of their paper. However, as I thought might be the case in my previous comments, I believe the case-mix analysis is not sensitive enough to make a valid assessment of a region’s complexity and performance as it is done at too high a level of aggregation: the ward or specialty. There can be considerable differences in case-mix within a ward that can be masked when using ward as a covariate to adjust for measures such as LOS or bed turnover.  For example, a region that has a high rate of referrals from other regions may appear to have low performance as a result of receiving a high proportion of complex cases relative to other regions. This would not be sufficiently captured by using ward as the adjustment variable. A patient admitted to the cardiac ward in region A with a mild heart attack would be equivalent to one admitted to region B with a more severe heart attack exacerbated by comorbidities such as diabetes. As documented in the literature on risk adjustment, to capture these differences among regions would require case-mix analysis at a DRG level at minimum and preferably with severity adjustment as well.

In addition, the measure of performance, bed occupancy rate, can be affected by other factors such as substitutability (e.g. an obstetrics patient can’t be put in infectious disease unit and vice-versa), regional differences in demand patterns (e.g. resort areas have more fluctuation over the year), and the size of a ward. The latter is probably the most important factor that needs to be considered as the larger the ward the smaller the coefficient of variation and thus the higher the occupancy rate it can maintain while still having enough beds to admit urgent patients. Without using ward size (or at least individual hospital size) occupancy rate comparisons can be very misleading.

Finally, there are still a few grammatical or word choice errors:

Line 108 – I believe the authors mean “most up to date” rather than “most outdated”

Line 209 – “institutions” is the correct word, not “institutes”

Line 185 – “reacted to this pandemic period” is stated; however, the data in the next section are from a period prior to the pandemic

Line 195 – “data” is a plural term so the verb should be “are”

Reviewer #2: (No Response)

7. PLOS authors have the option to publish the peer review history of their article (what does this mean?). If published, this will include your full peer review and any attached files.

Reviewer #1: No

Reviewer #2: **Yes: **Dr. A Peter

---

## [Editor Report · Acceptance letter]

11 Sep 2020

PONE-D-20-09436R1 

The efficiency in the ordinary hospital bed management in Italy: an in-depth analysis of intensive care unit in the areas affected by COVID-19 before the outbreak  

Dear Dr. Pecoraro:

I'm pleased to inform you that your manuscript has been deemed suitable for publication in PLOS ONE. Congratulations! Your manuscript is now with our production department. 

Kind regards, 

on behalf of

Professor Maria Michela Gianino 

Academic Editor

PLOS ONE